

# Correlation between preoperative uric acid levels and lymph node metastases in patients with papillary thyroid cancer: a retrospective study

ShaoKun Sun, Qin Zhou and Tao Hu

Department of Thyroid Surgery, The First People's Hospital of Kunshan, Suzhou, Jiangsu, China

## ABSTRACT

**Background**. In the past decade, numerous studies have highlighted a notable correlation between the prognosis of various cancers and uric acid (UA). Nevertheless, scientific literature regarding the association between central lymph node metastasis (CLNM) and UA remains lacking. This study sought to examine the association between UA and the likelihood of CLNM in individuals with papillary thyroid cancer (PTC) and to determine the UA threshold.

**Methods**. Three hundred and seventy individuals with confirmed PTC who underwent surgery at the First People's Hospital of Kunshan from October 2018 to July 2024 were included in this study. Initially, UA levels were measured, and the incidence of CLNM in PTC patients was assessed through postoperative pathology. Propensity score-matched (PSM) analysis was used to match patients. Multivariate logistic regression and a two-piecewise linear regression model were utilized to analyze the relationship between UA and CLNM and identify the threshold effect of UA concentrations on CLNM.

**Results**. The correlation between UA and CLNM in PTC patients was found to be positive. Upon adjusting for various confounding factors, it was determined that, relative to the referential level, the odds ratio for CLNM was 4.01 at peak UA levels. After PSM analysis, the odds ratio for CLNM was 5.23 at the highest levels to the lowest levels. A non-linear relationship between UA and CLNM emerged following adjustment for potential confounding factors. The study identified 219.20 µmol/L as the UA threshold, serving as the optimal inflection point. The effect sizes and confidence intervals on both sides of the inflection point were $-4.43$ ($-9.69$–$0.84$) and $3.78$ ($1.70$–$5.85$), respectively.

**Conclusion**. The study concludes that the association between UA and CLNM is non-linear. A positive relationship between UA and CLNM was observed when UA levels exceeded 219.20 µmol/L.

# INTRODUCTION

Papillary thyroid cancer (PTC), the predominant variant of thyroid malignancy, has shown an increasing incidence annually (*Miranda-Filho et al., 2021*). Despite the generally favorable prognosis for most PTC patients, early central lymph node metastases (CLNM) are common, which elevate the chance of local recurrence post-surgery (*Ye et al.,*

Corresponding author
Tao Hu, ht1801@Sina.com

*2023*). Additionally, the presence of CLNM in PTC patients is linked to a markedly elevated likelihood of mortality, distant metastases, and recurrence (*Wang & Ganly, 2016*). Consequently, therapeutic central lymph node dissection (PCLND) is advised for individuals exhibiting clinically positive central lymph nodes (cN1). Conversely, the necessity of prophylactic cervical lymph node dissection in PTC individuals with clinically negative central lymph nodes (cN0) remains contentious (*Haugen et al., 2015*; *Pavlidis & Pavlidis, 2023*). PCLND in cN0 PTC patients leads to prolonged disease-free survival and reduces local recurrence and metastasis in patients after surgery, but it can also elevate the occurrence of postoperative complications, such as recurrent laryngeal nerve injury and permanent hypoparathyroidism (*Dismukes et al., 2021*). Many patients may be in a compromised physical or psychological state following surgery, and are thus more likely to express more heightened regret compared to those who undergo active surveillance (*Li et al., 2025*). Presently, the preoperative diagnosis of CLNM predominantly depends on ultrasound; however, studies have indicated that a mere 33% of CLNMs can be accurately predicted using a single ultrasound (*Zhao & Li, 2019*). Therefore, a more convenient and efficient predictor of CLNM in individuals with PTC is needed. According to previous studies, uric acid (UA) can be a potential predictor of progression in lung and rectal cancers (*Yuan et al., 2016*; *Sayan et al., 2023*). However, minimal research has been conducted on the relationship between UA levels and CLNM in PTC.

Recent investigations into serum biomarkers have opened new avenues for cancer progression prediction. Of particular interest is UA, the end result of purine catabolism. Recent studies have increasingly associated elevated UA levels with metabolic syndrome (*Bowden, Richardson & Richardson, 2022*), which has emerged as a significant public health issue and has been linked to various malignancies, including thyroid cancer (TC) (*Park et al., 2020*). Metabolic syndrome, a cluster of conditions closely related to high UA, was also found to be a risk factor for CLNM, larger tumors, and later stage PTC (*Song et al., 2022*). UA crystals activate NLRP3 inflammatory vesicles, leading to the release of pro-inflammatory cytokines such as interleukin-1β and interleukin-18, which triggers a localized inflammatory response (*Martinon, 2010*). Chronic inflammation within the tumor microenvironment is known to significantly impact cancer progression and immunity (*Xie et al., 2020*). There is also evidence from previous studies that chronic inflammation may contribute to lymphatic invasion in PTC (*Kabasawa et al., 2021*).

Through its significant role in linking metabolic syndrome with cancer and its biological characterization of cancer progression, we hypothesized that there is a relationship between UA and lymph node metastasis in PTC. Clarifying whether UA influences CLNM in PTC patients is essential. However, limited investigations have examined the link between UA levels and CLNM risk. This research seeks to explore the correlation between UA and CLNM in PTC patients to find a more economical and convenient test to assist in the assessment of CLNM in patients with PTC.

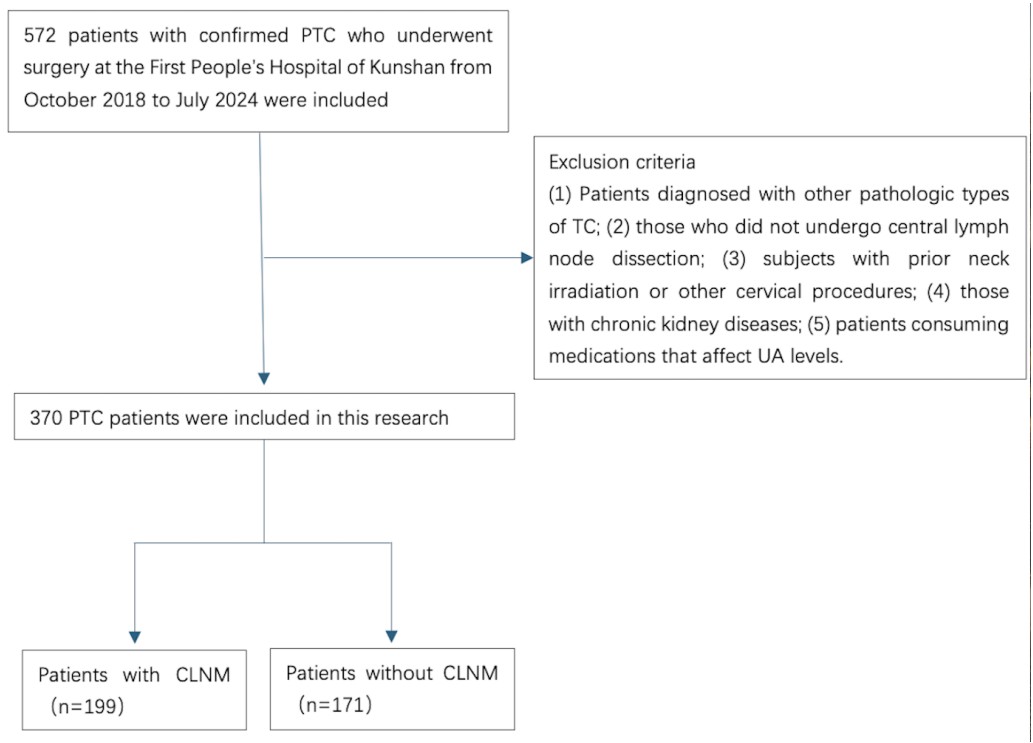

**Figure 1  A flowchart of the inclusion and exclusion criteria for the study.**

## MATERIALS & METHODS

### Patients

This research included 370 individuals with confirmed PTC who were treated at the Thyroid Surgery Department of the First People's Hospital of Kunshan between October 2018 and July 2024. Each patient underwent comprehensive treatment and examination at this institution. All surgical procedures were carried out by surgeons with more than 10 years of surgical expertise at our medical center. Patients were categorized into CLNM and non-CLNM cohorts based on the presence or absence of CLNM.

The selection criteria included: (1) individuals undergoing their initial thyroid surgery; (2) those with confirmed postoperative pathology; and (3) patients possessing complete clinical and ultrasound data.

The disqualification criteria comprised: (1) patients diagnosed with other pathologic types of TC; (2) those who did not undergo central lymph node dissection; (3) subjects with prior neck irradiation or other cervical procedures; (4) those with chronic kidney diseases; and (5) patients consuming medications that affect UA levels (Fig. 1).

### Data collection

Clinical information was collected, including UA (μmol/L), body mass index (BMI), creatinine (Cr) (μmol/L), triglyceride (TG) (mmol/L), low density lipoprotein cholesterol (LDL-C) (mmol/L), GLU (mmol/L), FT3 (pmol/L), FT4 (pmol/L), TSH (mIu/L) age, sex,
maximum tumor diameter, focality (unifocal or multifocal), extrathyroidal infiltration, tumor location (left, right, isthmus, bilobed), history of Hashimoto's thyroiditis (HT), BRAF gene mutation (V600E), and presence of CLNM in pathology, along with preoperative ultrasonographic characteristics like margin, shape, microcalcifications, and internal echogenicity.

Ultrasound scanning of the thyroid gland and central lymph nodes was conducted for all patients within 1 month before surgery. Experienced sonographers performed ultrasound examinations and documented the imaging features of thyroid tumors and central lymph nodes.

Blood samples were collected from all individuals between 6 and 7 AM after fasting for at least 8 h the previous night. Levels of UA, Cr, TG, LDL-C, GLU, FT3, FT4, and TSH in the blood were examined utilizing conventional laboratory techniques, which were conducted by a biochemistry lab technician at the First People's Hospital of Kunshan.

## Ethics statement

The ethics committee of the First People's Hospital of Kunshan granted approval for this retrospective study, which was classified as minimal risk, and a waiver of informed consent was obtained (Approval number: 2024-03-042-H00-K01). All methods used in this study were conducted in strict accordance with the ethical principles stated in the Declaration of Helsinki.

## Statistical analysis

Continuous variables were represented by weighted mean and standard deviation (SD), while categorical variables were expressed as frequency and weighted percentage. The distributions of CLNM and serum concentrations of UA were examined. Given the strong left skewness of UA data, a natural logarithmic transformation (Ln UA) was applied for the analysis. To reduce the potential impact of covariates (age, sex, and BMI), a PSM analysis was conducted. PSM was performed using the "nearest" method at a ratio of 1:1. One-way ANOVA, Kruskal-Wallis H tests, and chi-squared tests were utilized to assess statistical disparities between groups, and tests were selected based on data distribution and type. To evaluate the stability of LnUA's influence on CLNM across different regression models, LnUA values were divided into three cohorts on the basis of tertiles. LnUA was incorporated into the models both as a continuous variable and categorical variable (tertile grouping). The robustness of the association between LnUA and CLNM was further tested using three regression models, adjusting for confounding factors progressively: Model 1 included no covariate adjustments; Model 2 accounted for age and sex; and Model 3 accounted for all covariates. A generalized additive model was employed to detect non-linear relationships. In the presence of a non-linear relationship, a two-piecewise linear regression model was developed to estimate the threshold effect of LnUA on CLNM, as indicated by the smoothing plot. When the relationship between LnUA and CLNM was evident in the smoothed curve, a recursive method was employed to automatically identify the inflection point, which was subsequently applied in the following examinations.

All analyses were executed utilizing the R software (http://www.R-project.org, The R Foundation) and EmpowerStates (http://www.empowerstates.com, X & Y Solutions, Inc., Boston, MA). Statistical significance was established at $P < 0.05$.

## RESULTS

### Baseline characteristics of participants

This study included 370 participants, comprised of 43.78% males, with an average age of $43.12 \pm 11.71$ years. The average UA level was $321.79 \pm 96.44$ µmol/L. CLNM was present in 53.78% of individuals. Baseline characteristics are depicted in Table 1. Patients in the middle UA level group demonstrated a significantly higher incidence of CLNM than the low UA level group. Furthermore, the rate of CLNM in the high UA level group was significantly higher than that in the middle and low UA level groups.

### Association between the UA and CLNM before and after PSM

The findings from the univariate analysis are presented in Table 2. The results revealed that UA, BMI, sex, age, tumor size, extrathyroidal infiltration, HT, and microcalcifications were correlated with CLNM. Conversely, TG, LDL-C, Cr, GLU, FT3, FT4, TSH, focality, tumor location, BRAF V600E, margin, shape, and echogenicity were not associated with CLNM. After performing 1:1 PSM with age, sex, and BMI as covariates, 110 patients with CLNM and 110 patients without CLNM matched. UA, tumor size, extrathyroidal infiltration, and HT remained relevant factors in CLNM (Table 3).

Three regression models were constructed to investigate the independent impact of UA on CLNM before and after PSM, as shown in Tables 4 and 5. Both non-adjusted and adjusted models are displayed. In Model 1, LnUA exhibited a positive correlation with CLNM (odds ratio (OR) = 4.69, 95% confidence interval (CI) [2.22–9.91], $P < 0.0001$). The results remained consistent in Model 2 (adjusted for age and sex) and Model 3 (adjusted for all covariates) (OR = 7.55, 95% CI [2.71–21.07], $P = 0.0001$; OR = 7.77, 95% CI [2.04–29.65], $P = 0.0027$). After PSM, LnUA still showed a significant correlation with CLNM in the three models (OR = 4.37, 95% CI [1.73–11.04], $P = 0.0018$; OR = 8.22, 95% CI [2.44–27.67], $P = 0.0007$; OR = 9.65, 95% CI [1.80–51.86], $P = 0.0082$). For sensitivity analysis, LnUA was also treated as a categorical variable. Model 3 indicated that, in comparison to the low cohort, the high cohort had markedly elevated CLNM (OR = 4.01, 95% CI [1.71–9.42], $P = 0.0014$). However, there were no significant differences between the middle UA level group and the low UA level group ($P = 0.2335$). In the matched CLNM and non-CLNM groups, the difference between each UA level group was statistically significant.

### Subgroup analysis of the association between UA and the number of CLNM before and after PSM

Based on their number of lymph node metastases, patients were divided into two groups. Table 6 shows that the difference of the UA level between the two groups did not reach statistical significance ($P = 0.533$). Additionally, the difference did not show statistical significance after PSM ($P = 0.656$) (Table 7).

**Table 1   Baseline characteristics of participants.**

| Ln UA | Low (UA < 272 μmol/L) | Middle (UA: 272–344 μmol /L) | High (UA > 344 μmol/L) | P-value |
|---|---|---|---|---|
| N | 122 | 124 | 124 | |
| TG | 1.44 ± 0.90 | 1.56 ± 0.91 | 2.11 ± 1.22[a b] | <0.001 |
| LDL-C | 2.62 ± 0.74 | 2.76 ± 0.83 | 3.04 ± 0.78[a b] | <0.001 |
| BMI | 23.19 ± 1.56 | 23.68 ± 1.15[a] | 25.45 ± 1.51[a b] | <0.001 |
| Cr | 74.24 ± 14.09 | 76.75 ± 15.48 | 112.06 ± 45.36[a b] | <0.001 |
| GLU | 5.16 ± 0.79 | 5.27 ± 1.02 | 5.38 ± 0.75[a] | 0.118 |
| FT3 | 4.38 ± 0.66 | 4.45 ± 0.73 | 4.56 ± 1.25 | 0.313 |
| FT4 | 13.29 ± 1.85 | 13.15 ± 1.83 | 13.32 ± 2.43 | 0.782 |
| TSH | 1.75 ± 1.03 | 1.84 ± 1.26 | 1.71 ± 1.10 | 0.653 |
| CLNM | | | | <0.001 |
| No | 73 (59.84%) | 57 (45.97%)[a] | 41 (33.06%)[a b] | |
| Yes | 49 (40.16%) | 67 (54.03%) | 83 (66.94%) | |
| Sex | | | | 0.098 |
| Male | 49 (40.16%) | 64 (51.61%) | 49 (39.52%) | |
| Female | 73 (59.84%) | 60 (48.39%) | 75 (60.48%) | |
| Braf | | | | 0.500 |
| No | 29 (23.77%) | 25 (20.16%) | 22 (17.74%) | |
| Yes | 93 (76.23%) | 99 (79.84%) | 102 (82.26%) | |
| Age (years) | | | | 0.521 |
| <55 | 97 (79.51%) | 93 (75.00%) | 100 (80.65%) | |
| ≥55 | 25 (20.49%) | 31 (25.00%) | 24 (19.35%) | |
| Tumor size (cm) | | | | 0.616 |
| ≤0.5 | 100 (81.97%) | 96 (77.42%) | 101 (81.45%) | |
| >0.5 | 22 (18.03%) | 28 (22.58%) | 23 (18.55%) | |
| Focality | | | | 0.276 |
| Unifocal | 98 (80.33%) | 89 (71.77%) | 92 (74.19%) | |
| Multifocal | 24 (19.67%) | 35 (28.23%) | 32 (25.81%) | |
| Extrathyroidal infiltration | | | | 0.093 |
| No | 116 (95.08%) | 112 (90.32%) | 108 (87.10%)[a] | |
| Yes | 6 (4.92%) | 12 (9.68%) | 16 (12.90%) | |
| Hashimoto's thyroiditis | | | | 0.910 |
| No | 67 (54.92%) | 71 (57.26%) | 68 (54.84%) | |
| Yes | 55 (45.08%) | 53 (42.74%) | 56 (45.16%) | |
| Tumor location | | | | 0.119 |
| Left | 19 (15.57%) | 22 (17.74%) | 23 (18.55%) | |
| Right | 58 (47.54%) | 58 (46.77%) | 71 (57.26%) | |
| Isthmus | 38 (31.15%) | 30 (24.19%) | 20 (16.13%) | |
| Bilobed | 7 (5.74%) | 14 (11.29%) | 10 (8.06%) | |

**Table 1** (*continued*)

| Ln UA | Low (UA < 272 μmol/L) | Middle (UA: 272–344 μmol/L) | High (UA > 344 μmol/L) | *P*-value |
|---|---|---|---|---|
| Margin | | | | 0.766 |
|    Smooth | 37 (30.33%) | 43 (34.68%) | 40 (32.26%) | |
|    Ill-defined | 85 (69.67%) | 81 (65.32%) | 84 (67.74%) | |
| Shape | | | | 0.216 |
|    Oval | 35 (28.69%) | 24 (19.35%) | 28 (22.58%) | |
|    Taller-than-wide | 87 (71.31%) | 100 (80.65%) | 96 (77.42%) | |
| Echogenicity | | | | 0.070 |
|    Hypo echogenicity | 120 (98.36%) | 114 (91.94%)[a] | 117 (94.35%) | |
|    Iso/Hyperechogenicity | 2 (1.64%) | 10 (8.06%) | 7 (5.65%) | |
| Microcalcification | | | | 0.090 |
|    No | 67 (54.92%) | 62 (50.00%) | 51 (41.13%)[a] | |
|    Yes | 55 (45.08%) | 62 (50.00%) | 73 (58.87%) | |

**Notes.**
[a] $P < 0.05$ compared to the low group.
[b] $P < 0.05$ compared to the middle group.

## Analyses of a non-linear relationship before and after PSM

The investigation revealed a non-linear connection between LnUA and CLNM after adjusting for all covariates before and after PSM (Figs. 2 and 3). Utilizing a two-piecewise linear regression model, the inflection point was calculated to be LnUA = 5.39 (UA = 219.20 μmol/L). To the right of the inflection point, the impact magnitude was 2.89 (95% CI [1.33–4.45]) with $P = 0.0003$ (Table 8), and 3.78 (95% CI [1.70–5.85]) with $P = 0.0004$ after PSM (Table 9).

## DISCUSSION

To the best of our understanding, this research represents the initial exploration of the connection between UA and CLNM in PTC patients. Our hospital-based cohort investigation demonstrated that UA levels are linked to the likelihood of CLNM in PTC.

CLNM is frequently observed in PTC patients, and even those with cN0 PTC are often diagnosed with CLNM postoperatively (*Ma et al., 2023*). Currently, PCLND is recommended for cN1a and cN1b PTC patients. However, the necessity of lymph node removal in cN0 PTC cases continues to be a topic of discussion (*Haugen et al., 2015*; *Pavlidis & Pavlidis, 2023*). On the basis of expert consensus in China, PCLND may be conducted without jeopardizing the recurrent laryngeal nerve or parathyroid glands (*Zhao et al., 2017*). In contrast, the 2022 National Comprehensive Cancer Network guidelines advise against PCLND for cN0 individuals with T1 or T2 stage who lack high-risk factors for lymph node metastasis (*Haddad et al., 2022*). The sensitivity of preoperative detection methods, such as ultrasound, remains inadequate (*Zhao & Li, 2019*). Considering that the presence of preoperative CLNM influences surgical scope, it is crucial to determine a new predictor of CLNM in individuals with PTC and screen more patients with high risk of lymph node metastasis in order to assist clinicians in developing surgical plans.

**Table 2   Findings from the univariate analysis for factors related to CLNM.**

| Variable | CLNM(−) | CLNM(+) | P-value |
|---|---|---|---|
| N | 171 | 199 | |
| TG | 1.66 ± 1.05 | 1.75 ± 1.07 | 0.422 |
| LDL-C | 2.75 ± 0.87 | 2.86 ± 0.74 | 0.187 |
| BMI | 23.89 ± 1.67 | 24.30 ± 1.73 | 0.023 |
| Cr | 84.82 ± 32.16 | 90.28 ± 34.72 | 0.121 |
| GLU | 5.27 ± 0.82 | 5.27 ± 0.90 | 0.980 |
| FT3 | 4.42 ± 0.74 | 4.50 ± 1.05 | 0.401 |
| FT4 | 13.18 ± 1.70 | 13.31 ± 2.32 | 0.562 |
| TSH | 1.79 ± 1.13 | 1.74 ± 1.14 | 0.670 |
| Ln UA | 5.66 ± 0.28 | 5.79 ± 0.29 | <0.001 |
| Ln UA tertile | | | <0.001 |
|    Low | 73 (42.69%) | 49 (24.62%) | |
|    Middle | 57 (33.33%) | 67 (33.67%) | |
|    High | 41 (23.98%) | 83 (41.71%) | |
| Sex | | | <0.001 |
|    Male | 38 (22.22%) | 124 (62.31%) | |
|    Female | 133 (77.78%) | 75 (37.69%) | |
| Braf | | | 0.077 |
|    No | 42 (24.56%) | 34 (17.09%) | |
|    Yes | 129 (75.44%) | 165 (82.91%) | |
| Age (years) | | | 0.003 |
|    <55 | 122 (71.35%) | 168 (84.42%) | |
|    ≥55 | 49 (28.65%) | 31 (15.58%) | |
| Tumor size (cm) | | | <0.001 |
|    ≤0.5 | 154 (90.06%) | 143 (71.86%) | |
|    >0.5 | 17 (9.94%) | 56 (28.14%) | |
| Focality | | | 0.638 |
|    Unifocal | 127 (74.27%) | 152 (76.38%) | |
|    Multifocal | 44 (25.73%) | 47 (23.62%) | |
| Extrathyroidal infiltration | | | 0.019 |
|    No | 162 (94.74%) | 174 (87.44%) | |
|    Yes | 9 (5.26%) | 25 (12.56%) | |
| Hashimoto's thyroiditis | | | 0.033 |
|    No | 85 (49.71%) | 121 (60.80%) | |
|    Yes | 86 (50.29%) | 78 (39.20%) | |
| Tumor location | | | 0.208 |
|    Left | 24 (14.04%) | 40 (20.10%) | |
|    Right | 88 (51.46%) | 99 (49.75%) | |
|    Isthmus | 47 (27.49%) | 41 (20.60%) | |
|    Bilobed | 12 (7.02%) | 19 (9.55%) | |

**Table 2** (*continued*)

| Variable | CLNM(-) | CLNM(+) | *P*-value |
|---|---|---|---|
| Margin | | | 0.584 |
|   Smooth | 53 (30.99%) | 67 (33.67%) | |
|   Ill-defined | 118 (69.01%) | 132 (66.33%) | |
| Shape | | | 0.660 |
|   Oval | 42 (24.56%) | 45 (22.61%) | |
|   Taller-than-wide | 129 (75.44%) | 154 (77.39%) | |
| Echogenicity | | | 0.196 |
|   Hypo echogenicity | 165 (96.49%) | 186 (93.47%) | |
|   Iso/Hyperechogenicity | 6 (3.51%) | 13 (6.53%) | |
| Microcalcification | | | 0.004 |
|   No | 97 (56.73%) | 83 (41.71%) | |
|   Yes | 74 (43.27%) | 116 (58.29%) | |

Serum risk factors like UA may be less evident than traditional risk indicators such as tumor size, extrathyroidal infiltration, HT, age, and sex. Our study revealed a significant increase in CLNM risk at higher UA levels, particularly when UA exceeds 219.20 μmol/L. The highest UA levels were associated with a 5.23-fold heightened risk of CLNM in comparison to the baseline cohort ($P < 0.05$). The threshold of UA can be used as a potential biomarker to predict the risk of lymph node metastasis in patients with PTC. Identifying this threshold provides clinicians with a simple and easily detectable indicator to assess a patient's risk of metastasis preoperatively and thus optimize treatment options. Additionally, in the treatment of PTC, the extent and necessity of lymph node dissection have always been an important part of clinical decision-making. By testing UA levels, high-risk patients can be more accurately identified, thus avoiding unnecessary overtreatment or the omission of these patients.

Serum UA has been identified as a predictive factor for the prognosis of various cancers. A study identified a J-shaped connection between UA levels and breast cancer risk, with 3.6 mg/dl as the UA threshold (*Fan, Sun & Yin, 2023*). Elevated UA levels have been strongly correlated with both cancer development and increased cancer mortality, as demonstrated by a meta-analysis of 5 studies on cancer incidence and twelve on cancer mortality (*Yan et al., 2015*). *Hayashi et al. (2021)* analyzed 256 patients with surgically resected hepatocellular carcinoma and discovered that recurrence-free survival was markedly lower in individuals with elevated UA levels compared to those with low UA levels. High UA was identified as a substantial risk factor for recurrences of hepatocellular carcinoma. Evidence has also shown that preoperative UA is an independent prognostic predictor of overall survival in individuals with advanced cancer (*Zhao et al., 2019*). Additionally, it has been found that individuals with lymphatic metastasis for rectal cancer had higher serum UA concentrations than those without lymphatic metastasis (*Yuan et al., 2016*). *Sayan et al. (2023)* discovered that preoperative high UA levels were useful for predicting lymph node metastases in lung cancers treated by video-assisted thoracic surgery.

Although UA levels are not a defined component of metabolic syndrome, they are closely related. Lowering UA levels can halt or reverse metabolic syndrome (*Son et al., 2022*), as

**Table 3  Findings from the univariate analysis for factors related to CLNM after PSM.**

| Variable | CLNM(-) | CLNM(+) | *P*-value |
|---|---|---|---|
| N | 110 | 110 | |
| TG | 1.57 ± 0.98 | 1.72 ± 0.96 | 0.244 |
| LDL-C | 2.70 ± 0.82 | 2.84 ± 0.68 | 0.160 |
| BMI | 24.05 ± 1.78 | 24.21 ± 1.79 | 0.507 |
| Cr | 88.45 ± 36.50 | 93.66 ± 37.27 | 0.295 |
| GLU | 5.32 ± 0.89 | 5.38 ± 1.06 | 0.633 |
| FT3 | 4.47 ± 0.86 | 4.47 ± 0.61 | 0.988 |
| FT4 | 13.18 ± 1.86 | 13.18 ± 1.71 | 0.989 |
| TSH | 1.79 ± 1.10 | 1.63 ± 0.98 | 0.283 |
| Ln UA | 5.70 ± 0.30 | 5.83 ± 0.30 | 0.002 |
| Ln UA tertile | | | 0.002 |
| Low | 48 (43.64%) | 24 (21.82%) | |
| Middle | 33 (30.00%) | 41 (37.27%) | |
| High | 29 (26.36%) | 45 (40.91%) | |
| Sex | | | 0.888 |
| Male | 38 (34.55%) | 39 (35.45%) | |
| Female | 72 (65.45%) | 71 (64.55%) | |
| Braf | | | 0.141 |
| No | 28 (25.45%) | 19 (17.27%) | |
| Yes | 82 (74.55%) | 91 (82.73%) | |
| Age (years) | | | 1.000 |
| <55 | 89 (80.91%) | 89 (80.91%) | |
| ≥55 | 21 (19.09%) | 21 (19.09%) | |
| Tumor size (cm) | | | 0.024 |
| ≤0.5 | 97 (88.18%) | 84 (76.36%) | |
| >0.5 | 13 (11.82%) | 26 (23.64%) | |
| Focality | | | 1.000 |
| Unifocal | 84 (76.36%) | 84 (76.36%) | |
| Multifocal | 26 (23.64%) | 26 (23.64%) | |
| Extrathyroidal infiltration | | | 0.046 |
| No | 104 (94.55%) | 95 (86.36%) | |
| Yes | 6 (5.45%) | 15 (13.64%) | |
| Hashimoto's thyroiditis | | | 0.029 |
| No | 56 (50.91%) | 72 (65.45%) | |
| Yes | 54 (49.09%) | 38 (34.55%) | |
| Tumor location | | | 0.162 |
| Left | 15 (13.64%) | 23 (20.91%) | |
| Right | 55 (50.00%) | 50 (45.45%) | |
| Isthmus | 33 (30.00%) | 24 (21.82%) | |
| Bilobed | 7 (6.36%) | 13 (11.82%) | |

**Table 3** (*continued*)

| Variable | CLNM(-) | CLNM(+) | *P*-value |
|---|---|---|---|
| Margin | | | 0.386 |
|   Smooth | 38 (34.55%) | 32 (29.09%) | |
|   Ill-defined | 72 (65.45%) | 78 (70.91%) | |
| Shape | | | 0.094 |
|   Oval | 27 (24.55%) | 17 (15.45%) | |
|   Taller-than-wide | 83 (75.45%) | 93 (84.55%) | |
| Echogenicity | | | 0.757 |
|   Hypo echogenicity | 105 (95.45%) | 104 (94.55%) | |
|   Iso/Hyperechogenicity | 5 (4.55%) | 6 (5.45%) | |
| Microcalcification | | | 0.590 |
|   No | 56 (50.91%) | 52 (47.27%) | |
|   Yes | 54 (49.09%) | 58 (52.73%) | |

**Table 4** Correlation between UA and CLNM in diverse models.

| | Model 1 (OR, 95% CI, *P*) | Model 2 (OR, 95% CI, *P*) | Model 3 (OR, 95% CI, *P*) |
|---|---|---|---|
| Ln UA | 4.69 (2.22, 9.91) <0.0001 | 7.55 (2.71, 21.07) 0.0001 | 7.77 (2.04, 29.65) 0.0027 |
| Ln UA Tertile | | | |
|   Low | 1.0 | 1.0 | 1.0 |
|   Middle | 1.75 (1.06, 2.90) 0.0299 | 1.68 (0.94, 2.98) 0.0782 | 1.46 (0.78, 2.74) 0.2335 |
|   High | 3.02 (1.79, 5.08) <0.0001 | 4.34 (2.14, 8.80) <0.0001 | 4.01 (1.71, 9.42) 0.0014 |
| P for trend | <0.0001 | <0.0001 | 0.002 |

**Table 5** Correlation between UA and CLNM in diverse models after PSM.

| | Model 1 (OR, 95% CI, *P*) | Model 2 (OR, 95% CI, *P*) | Model 3 (OR, 95% CI, *P*) |
|---|---|---|---|
| Ln UA | 4.37 (1.73, 11.04) 0.0018 | 8.22 (2.44, 27.67) 0.0007 | 9.65 (1.80, 51.86) 0.0082 |
| Ln UA Tertile | | | |
|   Low | 1.0 | 1.0 | 1.0 |
|   Middle | 2.48 (1.27, 4.86) 0.0078 | 2.90 (1.43, 5.86) 0.0030 | 2.52 (1.15, 5.50) 0.0204 |
|   High | 3.10 (1.58, 6.11) 0.0010 | 4.97 (2.03, 12.16) 0.0004 | 5.23 (1.60, 17.11) 0.0062 |
| P for trend | 0.0010 | 0.0003 | 0.0033 |

UA has been linked to its onset. Metabolic syndrome encompasses a group of reversible and preventable disorders, encompassing central obesity, diabetes, hypertension, and dyslipidemia. A comprehensive, population-wide longitudinal investigation conducted in Korea revealed that subjects exhibiting metabolic syndrome faced an elevated likelihood of developing TC compared to those without the condition. The risk increased by 39% for individuals with all five metabolic syndrome components compared to those with none (*Park et al., 2020*). Consistent with these findings, the occurrence of TC per 10,000 person-years was significantly higher in subjects with metabolic syndrome compared to those without. Furthermore, the likelihood of developing TC increased markedly with the accumulation of metabolic syndrome elements, even in those with just one or two

**Table 6  Subgroup analysis of the association between UA and the number of CLNM.**

| Variable | 1-5 CLNM | >5 CLNM | P-value |
|---|---|---|---|
| N | 149 | 50 | |
| TG | 1.76 ± 1.12 | 1.73 ± 0.91 | 0.868 |
| LDL-C | 2.82 ± 0.72 | 2.98 ± 0.79 | 0.183 |
| BMI | 24.28 ± 1.82 | 24.37 ± 1.47 | 0.734 |
| Cr | 90.60 ± 35.31 | 89.32 ± 33.20 | 0.821 |
| GLU | 5.32 ± 0.95 | 5.11 ± 0.70 | 0.162 |
| FT3 | 4.48 ± 1.19 | 4.55 ± 0.47 | 0.680 |
| FT4 | 13.29 ± 2.42 | 13.35 ± 1.98 | 0.879 |
| TSH | 1.70 ± 1.13 | 1.87 ± 1.16 | 0.364 |
| Ln UA | 5.80 ± 0.31 | 5.77 ± 0.23 | 0.533 |
| Ln UA tertile | | | 0.081 |
|    Low | 52 (34.90%) | 14 (28.00%) | |
|    Middle | 43 (28.86%) | 23 (46.00%) | |
|    High | 54 (36.24%) | 13 (26.00%) | |
| Sex | | | 0.534 |
|    Male | 91 (61.07%) | 33 (66.00%) | |
|    Female | 58 (38.93%) | 17 (34.00%) | |
| Braf | | | 0.814 |
|    No | 26 (17.45%) | 8 (16.00%) | |
|    Yes | 123 (82.55%) | 42 (84.00%) | |
| Age (years) | | | 0.041 |
|    <55 | 121 (81.21%) | 47 (94.00%) | |
|    ≥55 | 28 (18.79%) | 3 (6.00%) | |
| Tumor size (cm) | | | <0.001 |
|    ≤0.5 | 118 (79.19%) | 25 (50.00%) | |
|    >0.5 | 31 (20.81%) | 25 (50.00%) | |
| Focality | | | 0.222 |
|    Unifocal | 117 (78.52%) | 35 (70.00%) | |
|    Multifocal | 32 (21.48%) | 15 (30.00%) | |
| Extrathyroidal infiltration | | | 0.185 |
|    No | 133 (89.26%) | 41 (82.00%) | |
|    Yes | 16 (10.74%) | 9 (18.00%) | |
| Hashimoto's thyroiditis | | | 0.593 |
|    No | 89 (59.73%) | 32 (64.00%) | |
|    Yes | 60 (40.27%) | 18 (36.00%) | |
| Tumor location | | | 0.601 |
|    Left | 30 (20.13%) | 10 (20.00%) | |
|    Right | 75 (50.34%) | 24 (48.00%) | |
|    Isthmus | 28 (18.79%) | 13 (26.00%) | |
|    Bilobed | 16 (10.74%) | 3 (6.00%) | |

**Table 6** (*continued*)

| Variable | 1-5 CLNM | >5 CLNM | *P*-value |
|---|---|---|---|
| Margin | | | 0.773 |
|   Smooth | 51 (34.23%) | 16 (32.00%) | |
|   Ill-defined | 98 (65.77%) | 34 (68.00%) | |
| Shape | | | 0.152 |
|   Oval | 30 (20.13%) | 15 (30.00%) | |
|   Taller-than-wide | 119 (79.87%) | 35 (70.00%) | |
| Echogenicity | | | 0.081 |
|   Hypo echogenicity | 142 (95.30%) | 44 (88.00%) | |
|   Iso/Hyperechogenicity | 7 (4.70%) | 6 (12.00%) | |
| Microcalcification | | | 0.110 |
|   No | 67 (44.97%) | 16 (32.00%) | |
|   Yes | 82 (55.03%) | 34 (68.00%) | |

(*Park, Cho & Yoon, 2022*). Prior research has demonstrated that metabolic syndrome is linked to PTC aggressiveness, such as tumor size > 1 cm and lymph node metastasis. This connection remained notable even when controlling for factors such as age, sex, TSH, and BMI (*Song et al., 2022*). Insulin resistance, a crucial element in metabolic syndrome's pathophysiology, can enhance proliferation, angiogenesis, cellular mobility, apoptosis, and DNA damage in cancer cells (*Oh et al., 2011*). *Zhao et al. (2022)* conducted a hospital-based case-control study and found that insulin resistance was markedly linked to the carcinogenesis and aggressiveness of PTC. Another study indicated that insulin resistance elevates the likelihood of recurrence in TC patients, linking it to a higher incidence of structural persistent disease at the final follow-up (*Pitoia et al., 2015*). Among PTC patients, a higher homeostasis model assessment of insulin resistance was associated with the multifocality of PTC, suggesting a correlation between insulin resistance and PTC aggressiveness (*Bae et al., 2016*). The possibility of progression in PTC may be attributable to chronic inflammation caused by metabolic syndrome. Chronic inflammation can promote tumor cell invasion and metastasis while affecting the tumor microenvironment (*Xie et al., 2020*). Elevated levels of insulin and insulin-like growth factor 1 in patients with metabolic syndrome can increase the invasion and metastasis of tumor cells through the activation of the PI3K/AKT/mTOR pathway (*Nwabo Kamdje et al., 2022*). Abnormally elevated blood lipids may provide tumor cells with more energy and biosynthetic raw materials, promoting their growth and metastasis (*Liu, Hu & Liu, 2022*).

Furthermore, UA demonstrates a potent ability to neutralize free radicals in human blood, serving as a natural antioxidant. The predominant antioxidant capacity in plasma can be attributed to the potent action of UA. By inhibiting the formation of oxygen radicals and lipid peroxidation, UA demonstrates significant anti-cancer properties. *Ames et al. (1981)* initially proposed that UA operates as a scavenger of singlet oxygen and hydroxyl radicals, thereby inhibiting lipid peroxidation in erythrocytes and providing substantial defense against human cancer. A large cohort study based on the general population revealed a strong association between UA levels exceeding 5.8 mg/dl and a reduced risk of cancer mortality (*Taghizadeh, Vonk & Boezen, 2014*). In line with this conclusion, research

**Table 7  Subgroup analysis of the association between UA and the number of CLNM after PSM.**

| Variable | 1-5 CLNM | >5 CLNM | *P*-value |
|---|---|---|---|
| N | 85 | 25 | |
| TG | 1.72 ± 0.95 | 1.73 ± 1.03 | 0.979 |
| LDL-C | 2.78 ± 0.66 | 3.05 ± 0.74 | 0.094 |
| BMI | 24.14 ± 1.82 | 24.43 ± 1.72 | 0.484 |
| Cr | 92.05 ± 36.22 | 99.16 ± 40.93 | 0.402 |
| GLU | 5.44 ± 1.12 | 5.21 ± 0.81 | 0.359 |
| FT3 | 4.42 ± 0.65 | 4.65 ± 0.38 | 0.097 |
| FT4 | 13.18 ± 1.83 | 13.16 ± 1.25 | 0.967 |
| TSH | 1.61 ± 0.94 | 1.71 ± 1.13 | 0.652 |
| Ln UA | 5.83 ± 0.32 | 5.86 ± 0.21 | 0.656 |
| Ln UA tertile | | | 0.499 |
|     Low | 31 (36.47%) | 6 (24.00%) | |
|     Middle | 27 (31.76%) | 9 (36.00%) | |
|     High | 27 (31.76%) | 10 (40.00%) | |
| Sex | | | 0.948 |
|     Male | 30 (35.29%) | 9 (36.00%) | |
|     Female | 55 (64.71%) | 16 (64.00%) | |
| Braf | | | 0.432 |
|     No | 16 (18.82%) | 3 (12.00%) | |
|     Yes | 69 (81.18%) | 22 (88.00%) | |
| Age (years) | | | 0.990 |
|     <55 | 64 (75.29%) | 25 (100.00%) | |
|     ≥55 | 21 (24.71%) | 0 (0.00%) | |
| Tumor size (cm) | | | 0.032 |
|     ≤0.5 | 69 (81.18%) | 15 (60.00%) | |
|     >0.5 | 16 (18.82%) | 10 (40.00%) | |
| Focality | | | 0.560 |
|     Unifocal | 66 (77.65%) | 18 (72.00%) | |
|     Multifocal | 19 (22.35%) | 7 (28.00%) | |
| Extrathyroidal infiltration | | | 0.786 |
|     No | 73 (85.88%) | 22 (88.00%) | |
|     Yes | 12 (14.12%) | 3 (12.00%) | |
| Hashimoto's thyroiditis | | | 0.435 |
|     No | 54 (63.53%) | 18 (72.00%) | |
|     Yes | 31 (36.47%) | 7 (28.00%) | |
| Tumor location | | | 0.360 |
|     Left | 20 (23.53%) | 3 (12.00%) | |
|     Right | 38 (44.71%) | 12 (48.00%) | |
|     Isthmus | 16 (18.82%) | 8 (32.00%) | |
|     Bilobed | 11 (12.94%) | 2 (8.00%) | |

**Table 7** (*continued*)

| Variable | 1-5 CLNM | >5 CLNM | *P*-value |
|---|---|---|---|
| Margin | | | 0.360 |
| Smooth | 27 (31.76%) | 5 (20.00%) | |
| Ill-defined | 58 (68.24%) | 20 (80.00%) | |
| Shape | | | 0.932 |
| Oval | 13 (15.29%) | 4 (16.00%) | |
| Taller-than-wide | 72 (84.71%) | 21 (84.00%) | |
| Echogenicity | | | 0.529 |
| Hypo echogenicity | 81 (95.29%) | 23 (92.00%) | |
| Iso/Hyperechogenicity | 4 (4.71%) | 2 (8.00%) | |
| Microcalcification | | | 0.086 |
| No | 44 (51.76%) | 8 (32.00%) | |
| Yes | 41 (48.24%) | 17 (68.00%) | |

indicated that breast cancer mortality rates were inversely correlated with UA levels (*Kühn et al., 2017*). However, a recent study discovered that the antioxidant capacity of UA is inferior to that of hydrophilic vitamin C or hydrophobic vitamin E, and its antioxidant effect is not as pronounced as previously believed (*Liu et al., 2019*). Contrary to the suggested antioxidant and protective effects of UA against cancer, *Strasak et al. (2007)* proposed that elevated UA concentrations might be linked to outcomes indicative of more severe prognoses. Given the controversy regarding the connection between UA and cancer progression, as well lymph node metastasis being considered a marker of this progression, the purpose of this study was to identify the association between UA and the likelihood of CLNM in individuals with PTC.

While the precise mechanisms by which UA is associated with cancer or its aggressiveness remain elusive, accumulating evidence indicates that UA may contribute to carcinogenesis (*Yan et al., 2015*). UA, a pro-inflammatory substance, has been associated with a higher incidence of cancer and cardiovascular diseases and a poorer prognosis. The pro-inflammatory effects of UA are exerted in two primary ways: through monosodium urate (MSU) crystals and soluble factors. The inflammation induced by MSU is primarily mediated by elements of the innate immune mechanism. When MSU activates mononuclear phagocytic cells, it initiates an intricate inflammatory sequence, leading to the release of cytokines and chemokines like IL-1β, TNF, IL-6, and IL-8. MSU has been shown to induce cytokine production partly through receptors on the plasma membrane, including toll-like receptors (*Martinon, 2010*). Conversely, soluble UA penetrates cells to activate MAP kinases p38 and ERK1/2, which subsequently activate NF-KB and induce the expression of inflammatory mediators like MCP-1 and C-reactive protein (*Sautin et al., 2007*). Recent research has demonstrated that the development and progression of PTC are intimately linked to the immune-inflammatory response within the tumor microenvironment (*Xie et al., 2020*). Blood inflammatory markers, acting as circulating immune markers, can reflect the body's immune status and have the potential to mirror the nature of thyroid nodules and predict TC prognosis (*Huang et al., 2022*). The platelet-to-lymphocyte count ratio (PLR) is broadly acknowledged as a marker of poor prognosis in

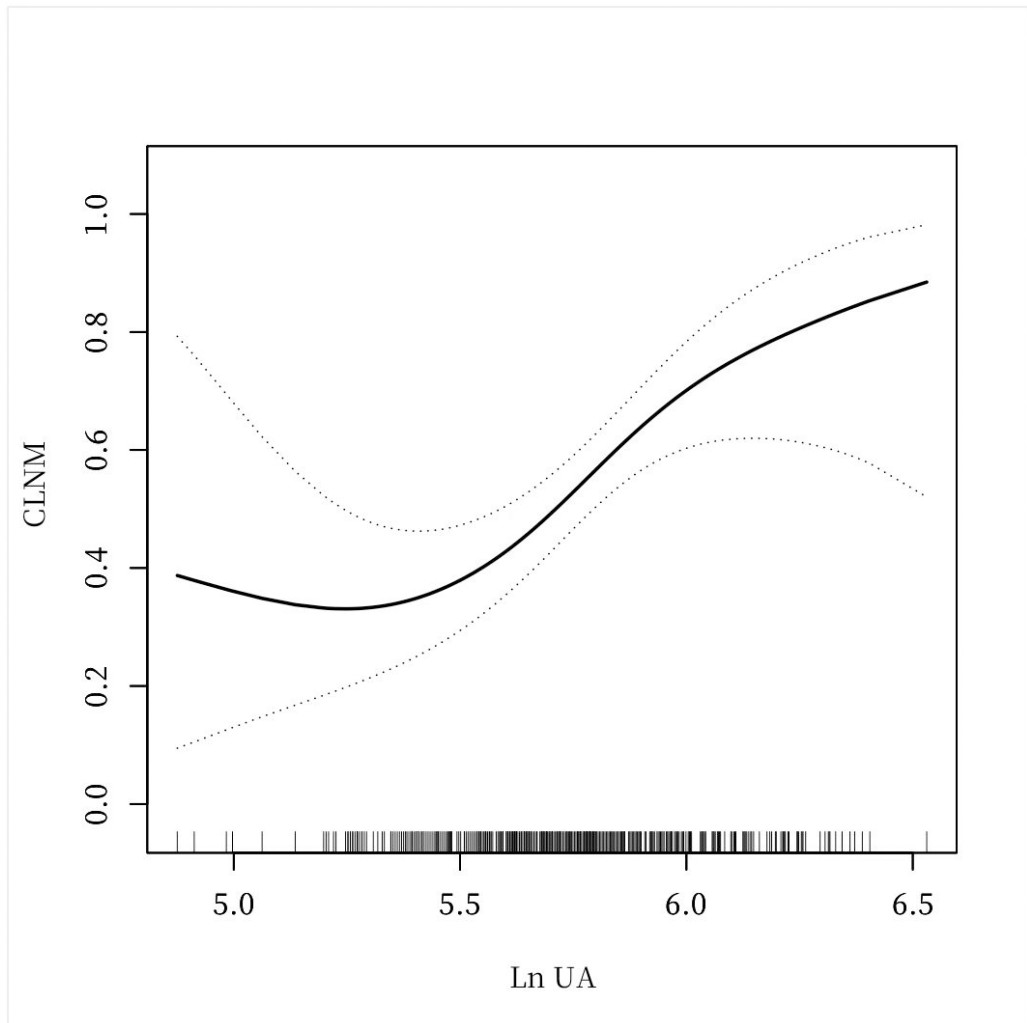

**Figure 2    A non-linear connection between UA levels and CLNM in PTC patients.**

various malignancies. *Kim et al. (2015)* observed that PTC patients with higher preoperative PLR exhibited a significantly increased incidence of lymph node metastasis. Beyond inflammatory stress, reactive oxygen species (ROS) generated by UA are considered a critical factor in cancer development. ROS are believed to play a dual role in regulating tumor cell signaling pathways (*Snezhkina et al., 2019*). A slight increase in ROS can promote cancer occurrence, growth, and metastasis by activating signaling pathways related to angiogenesis, proliferation, survival, and metastasis (*Perillo et al., 2020*). Additionally, ROS can enhance cell growth, migration, and invasion in TC cells (*Kim et al., 2023*).

Previous studies have shown some correlation between UA and cancer progression. *Sayan et al. (2023)* analyzed 115 patients with non-small cell lung cancer and found a statistically significant correlation between lymph node metastasis and high UA levels. However, the study had a small sample size and patients were divided into only two groups based on UA levels in the statistical analysis. A research conducted by *Yuan et*

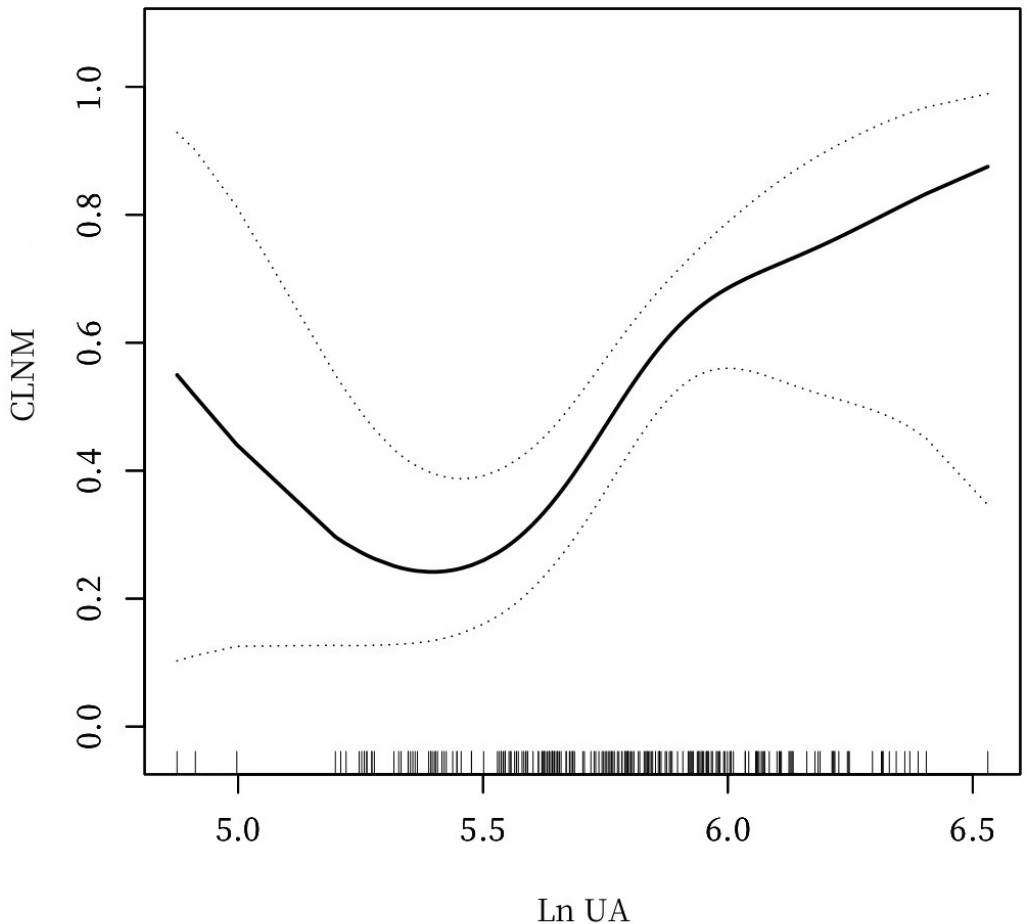

**Figure 3** **A non-linear connection between UA levels and CLNM in PTC patients after PSM.**

*al. (2016)* showed that serum UA may be a novel marker in assessing tumor metastasis in patients with rectal cancer. Since few clinical parameters related to disease severity were included in multiple regression analysis, the findings of the study require further validation. *Bozlak et al. (2016)* conducted a prospective cohort study and suggested that high UA levels were common in the malignancy group. However, the disease studied was cervical lymphadenopathy and the age of the study subjects was limited between 79.4 ± 46.7 months. Our study represents the first exploration of the connection between UA and CLNM in PTC patients and included 370 PTC patients. PSM analysis was used to match patients. Multivariate logistic regression and a two-piecewise linear regression model were used to analyze the relationship between UA and CLNM and identify the threshold effect of UA concentrations on CLNM.

The current investigation revealed that elevated UA levels surpassing 219.20 μmol/L were associated with an increased likelihood of CLNM occurrence. Consequently, individuals, particularly those with high-risk factors such as extrathyroidal infiltration, who exhibit elevated UA levels require thorough examination to evaluate lymph node status. To our

**Table 8  The threshold effect of LnUA level on CLNM.**

| Ln UA | OR (95% CI) | P- value |
|---|---|---|
| Inflection point | 5.39 | |
| <Inflection point | −3.01 (−7.37, 1.36) | 0.1777 |
| >Inflection point | 2.89 (1.33, 4.45) | 0.0003 |

**Table 9  The threshold effect of LnUA level on CLNM after PSM.**

| Ln UA | OR (95% CI) | P- value |
|---|---|---|
| Inflection point | 5.39 | |
| <Inflection point | −4.43 (−9.69, 0.84) | 0.0993 |
| >Inflection point | 3.78 (1.70, 5.85) | 0.0004 |

current understanding, this study is the first investigation to propose a possible association between UA and CLNM in PTC patients. The analysis revealed a non-linear relationship between LnUA and CLNM risk after adjusting for all other variables ($P < 0.05$), with a threshold for the CLNM risk curve identified at a UA level of 219.20 µmol/L, beyond which the risk escalated rapidly.

It must be acknowledged, however, that our study presents certain limitations. Primarily, the analytical cross-sectional design of this study offers restricted evidence for associating exposure with outcome, complicating the determination of causality. Additionally, the small sample size precludes the possibility of conducting stratified analyses. Furthermore, the single-center nature of this study restricts the applicability of our findings to a broader context. Consequently, future research employing a multi-institutional prospective design with a more extensive and diverse cohort will be crucial to confirm the reliability of our statistical approach.

## CONCLUSION

The association between LnUA and CLNM exhibits a non-linear pattern. A positive correlation between UA and CLNM is observed when UA levels exceed 219.20 µmol/L. UA may be a new predictor of CLNM in PTC patients that can help in the screening of more patients with high risk of lymph node metastasis and development of surgical plans for clinicians. It is imperative to conduct a more comprehensive prospective study to assess the precision and threshold of this relationship for predicting the risk of CLNM and elucidate the underlying pathogenic mechanisms.

## ACKNOWLEDGEMENTS

The authors express their deep gratitude to all contributors, including participants, surgeons, pathologists, sonographers, hospitals, and engineers, for their invaluable efforts in establishing the cohort and enabling the completion of this research. Their collaborative work has been pivotal to the success of the study.

### Funding
The authors received no funding for this work.

### Competing Interests
The authors declare there are no competing interests.

### Author Contributions
- ShaoKun Sun conceived and designed the experiments, performed the experiments, analyzed the data, prepared figures and/or tables, authored or reviewed drafts of the article, and approved the final draft.
- Qin Zhou performed the experiments, analyzed the data, prepared figures and/or tables, and approved the final draft.
- Tao Hu conceived and designed the experiments, analyzed the data, authored or reviewed drafts of the article, and approved the final draft.

### Human Ethics
The following information was supplied relating to ethical approvals (i.e., approving body and any reference numbers):

The Ethics Committee of The First People's Hospital of Kunshan (Ethical Application Ref: 2024-03-042-H00-K01).

### Data Availability
The raw measurements are available in the Supplementary File.

### Supplemental Information
Supplemental information for this article can be found online at http://dx.doi.org/10.7717/peerj.19410#supplemental-information.

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
