# Peer review of "Correlation between preoperative uric acid levels and lymph node metastases in patients with papillary thyroid cancer: a retrospective study"

_PeerJ, doi:10.7717/peerj.19410_

## Round 0.1 · original submission · Major Revisions

Dear authors,

Thank you for your submission. At this stage, your manuscript requires significant revisions.

You should pay particular attention to the fact that the correlations suggested are not supported by your evidence/have no clinical utility in real life practice. Thus, this clearly should be addressed and thought through.

Reviewer 1 ·

Basic reporting

247 attributed to the potent action of UA. By inhibiting the formation of oxygen radicals and lipid
248 peroxidation, UA demonstrates significant anti-cancer properties.

261 remain elusive, accumulating evidence indicates that UA may contribute to carcinogenesis 16.
262 UA, a pro-inflammatory substance, has been associated with a higher incidence of cancer and
263 cardiovascular diseases and a poorer prognosis


In the above lines you present clear contradicting information about the role of UA in carcinogenesis

Often difficult to follow along with text

Experimental design

158 UA, sex, age, tumor size, extrathyroidal infiltration, HT, and microcalcifications were correlated
159 with CLNM. Conversely, GLU, FT3, FT4, TSH, focality, tumor location, BRAF V600E, margin,
160 shape, and echogenicity were not associated with CLNM

Why did you choose to test for T3, T4, as it relates to central lymph node mets? Is there any evidence that you know of that supports a correlation between these levels and central lymph node mets?

Validity of the findings

I think it is difficult to argue any real significant relevance of UA level on central lymph node mets with Papillary thyroid cancer.

Even if you are suggesting any correlation in this case, you would need to make a case of why this matters. What would be the utility of testing patients with papillary thyroid cancer for Uric acid levels? Are you suggesting that UA can somehow be used as a surrogate endpoint for evaluation of cervical mets (i.e. prognostic factor)? If that is your argument, then you need to provide some evidence for that.

Similarly, if you are trying to argue a potential therapeutic target: (i.e. targeting UA If it contributes to tumor progression) you need to provide some evidence of something similar being done in the past.

Reviewer 2 ·

Basic reporting

This study investigated the correlation between preoperative serum uric acid levels and lymph node metastasis in patients with papillary thyroid carcinoma (PTC), and the selected topic has clinical practical value and the study design is basically reasonable. However, there are the following problems that need to be added and improved, and it is recommended to reconsider the study after making modifications in the following areas.

Experimental design

Methods
1. Insufficient control for confounding factors. It is better to correct for the effects of renal function indicators (e.g. eGFR) and metabolic syndrome-related parameters (BMI, blood lipids, etc.) on uric acid levels.
2. No subgroup analysis according to the number of CLNM
3. No information about lateral neck lymph node metastases
4. Add specific methods of testing UA (enzyme/HPLC?)

Validity of the findings

1. The current sample size is not valid enough, and selection bias may exist in single-centre studies.

Additional comments

Introduction
1.Need to strengthen the logical chain: existing research gaps → biological properties of uric acid → proposed research hypothesis

Discussion
1.No in-depth comparison with similar studies (e.g. relationship between uric acid and metastasis of other head and neck tumours).

Reviewer 3 ·

Basic reporting

This manuscript investigates the correlation between preoperative uric acid (UA) levels and central lymph node metastasis (CLNM) in patients with papillary thyroid cancer (PTC). The study provides novel insights into the potential role of UA as a biomarker for CLNM, which could have significant implications for clinical decision-making regarding prophylactic central lymph node dissection. The research is well-structured, with clear objectives, robust methodology, and comprehensive statistical analyses. However, there are several areas that could be improved to strengthen the study's validity and impact.

Experimental design

1. The authors have described the inclusion and exclusion criteria for participants as well as the data processing methods in the Methods section. Presenting this information in the form of a flowchart would significantly enhance the reader’s understanding of the study design and participant selection process.
2. In Section 2.1, the authors describe the characteristics of participants across different uric acid concentration groups. The p-values provided appear to reflect overall differences in the distribution of single features among the three groups, which does not necessarily indicate that there are no differences between each pair of groups. Moreover, since the authors aim to describe the relationship between different uric acid levels and lymph node metastasis, it would be beneficial to also present the differences in the distribution of participant characteristics between those with and without lymph node metastasis. This would help to clarify whether the presence of lymph node metastasis is potentially influenced by these covariates.

Validity of the findings

3. The authors discuss the association between uric acid levels and lymph node metastasis using multivariate Cox regression. Given the unclear differences in characteristics between patients with and without metastasis at each uric acid concentration, propensity score matching (PSM) of participants could be considered to reduce the potential impact of covariates.
4. The authors have identified a threshold for the association between uric acid and lymph node metastasis. It is necessary to further elaborate on the biological significance and potential clinical value of this threshold. The calculation of the threshold and the linear relationship may be influenced by the data, and thus a detailed explanation of its implications is warranted.

---

## Round 0.2 · Minor Revisions

Dear authors,

Please, clarify the remaining questions/comments.

Reviewer 2 ·

Basic reporting

Author's Response Comments were generally adequate and satisfactory.
However, the core issue of this paper that remains is that the proposed correlation between uric acid and PTC metastasis is slightly abrupt, and the introduction section needs to continue to be revised, not so much for the need for a large presentation of the biological properties of uric acid, but rather how to present your research hypothesis in a more reasonable manner based on the paper that have been presented.

Experimental design

NA

Validity of the findings

NA

Additional comments

Also, the possible mechanisms of metabolism and thyroid cancer metastasis need to be discussed in depth in the discussion section.

Reviewer 3 ·

Basic reporting

After the author's revisions, the study has conducted a good analysis of the collected data. The reliability of the article's conclusions has been significantly improved. In line 55 of the article, an attempt could be made to expand the description of the damage caused by the surgery to the patients (PMID: 40057484) to further enhance the clinical value of the article.

Experimental design

no comment

Validity of the findings

no comment

Additional comments

no comment

---

## Round 0.3 · accepted · Accept

Dear authors,

i am accepting your work for publication. Congratulations! Please, note that during the proofreading stage you will need to fix small typos and mistakes, such as in the following sentences eg. "Many patients may in impaired physical or psychological status after surgery, and likely to express more heightened regret than those ..", and ". The possible of metabolic syndrome and progression in PTC may attributable to chronic inflammation caused by metabolic syndrome. Chronic inflammation can promote tumor cell invasion and metastasis ..."